# Gross Motor Proficiency and Reading Abilities Among Chinese Primary School Students

**DOI:** 10.3390/bs15121613

**Published:** 2025-11-23

**Authors:** Tongtong Shao, Feng Lu, Dingzhou Liu, Hongfan Chen, Haomin Zhang

**Affiliations:** Faculty of Humanities and Social Sciences, City University of Macau, Avenida Padre Tomás Pereira, Taipa, Macau SAR 999078, China; h24092110425@cityu.edu.mo (T.S.); h24092110161@cityu.edu.mo (F.L.); h24092110331@cityu.edu.mo (D.L.); h24092110452@cityu.edu.mo (H.C.)

**Keywords:** gross motor proficiency, phonological awareness, morphological awareness, vocabulary knowledge, reading comprehension

## Abstract

The relation between motor skills and reading performance among young children has been explored in existing studies, but few of them focused on gross motor skills, and these demonstrated inconsistent findings. The current study aimed to examine the relationship between gross motor proficiency and reading abilities among Chinese primary school students (*N* = 107, mean age = 8.70 years). Participants completed measures of a non-verbal intelligence test, a gross motor proficiency test, and reading ability tests that assess their Chinese phonological awareness, Chinese morphological awareness, vocabulary knowledge, and reading comprehension. The results of correlational and regression analyses revealed a weak association between gross motor level and each component of reading achievement. Meanwhile, the correlation between gross motor proficiency and morphological awareness, as well as between gross motor proficiency and reading comprehension, did not reach statistical significance. In conclusion, the present study justified the negligible predictive power of gross motor proficiency on reading abilities among Chinese young students.

## 1. Introduction

### 1.1. Conceptualizing Motor Skills and Their Educational Relevance

Motor skills represent a complex constellation of internal processes that orchestrate bodily movement through intricate sensorimotor integration and cognitive processing ([9]). Within educational contexts, motor skills are conventionally categorized into gross and fine motor domains ([7]; [29]; [55]), both recognized as foundational elements in primary students’ holistic development ([46]). Gross motor skills engage large muscle groups to facilitate whole-body movements including balance maintenance, postural control, and coordinated actions such as running, jumping, and throwing ([8]; [21]). Conversely, fine motor skills encompass precision-based small-muscle activities requiring sophisticated hand-eye coordination, visuo-motor integration, and graphomotor competence essential for handwriting, drawing, and instrumental play ([4]; [9]; [38]). The theoretical linkage between motor functioning and cognitive development has garnered substantial scholarly attention ([24]; [47]). Within this conceptual framework, reading—as a complex cognitive endeavor ([11])—assumes particular significance due to its fundamental role in knowledge acquisition and academic advancement ([5]). This establishes the theoretical premise for investigating how motor proficiency might facilitate or correlate with reading development.

### 1.2. Evidentiary Patterns in Motor–Reading Relationships

Empirical investigations have yielded compelling evidence regarding fine motor skills’ association with reading achievement. Multiple studies consistently demonstrate that fine motor competencies, particularly graphomotor skills and handwriting fluency, significantly predict reading outcomes across diverse learner populations ([2]; [38]; [51]). This relationship appears robust enough that kindergarten fine motor capabilities, especially visual–motor integration, contribute unique variance in predicting subsequent reading proficiency in early elementary years ([5]; [50]). Contrasting sharply with the relative consensus surrounding fine motor skills, the gross motor–reading relationship presents a landscape marked by theoretical ambiguity and empirical contradictions. Systematic reviews and meta-analyses encapsulate this dissonance: while [31] ([31]) synthesized evidence supporting positive associations between gross motor proficiency and academic achievement, [55]’s ([55]) rigorous analysis of 78 studies concluded that gross motor skills demonstrated no significant correlation with reading performance. This fundamental discrepancy underscores a critical knowledge gap in our understanding of how different motor domains interact with reading development.

### 1.3. Methodological and Conceptual Challenges in Existing Literature

The empirical discordance regarding gross motor skills’ role in reading acquisition may stem from several methodological and conceptual factors. First, significant variation exists in how studies operationalize and measure gross motor proficiency—some investigations focus on composite scores ([32]), while others examine specific components like locomotor skills or object control ([56]). Second, developmental timing appears crucial, with gross motor skills potentially exerting differential influences across distinct educational stages ([33]). Perhaps most notably, the literature reveals intriguing demographic and contextual patterns that further complicate this relationship. [37] ([37]) documented a counterintuitive negative association between gross motor proficiency and reading achievement among Australian boys, suggesting potential gender-specific dynamics. Simultaneously, children with initial learning difficulties appeared to derive particular benefit from motor skill interventions, indicating possible compensatory mechanisms ([37]). These nuanced findings highlight the inadequacy of simplistic, unidirectional models and underscore the need for more sophisticated, contextually sensitive investigative approaches.

### 1.4. Linguistic and Cultural Considerations in Motor–Reading Relationships

A particularly salient limitation in current understanding concerns the overwhelming reliance on alphabetic language contexts in existing research. Chinese is a morphosyllabary language relying on the mapping principle of graph to word, which is distinct from the alphabetic languages, like English, that transform graph to phoneme ([13]; [41]). Meanwhile, Chinese morphology contains fewer alterations of inflectional and derivational affixes ([14]; [26]), whereas there is a productive derivational system in English, including inflection, derivation, and compounding ([43]; [59]). Additionally, phonological priming tends to contribute to English reading, but morphological awareness shows a stronger power in Chinese literacy acquisition ([27]; [34]; [42]). This indicates the great differences between Chinese and English reading mechanisms. The distinctive cognitive–linguistic demands of Chinese literacy acquisition—including character recognition, stroke sequencing, and spatial configuration—may engender qualitatively different relationships between motor competence and reading development compared to alphabetic systems. The graphomotor precision required for Chinese character writing, coupled with the visual–spatial processing involved in character recognition, suggests that fine motor skills might assume heightened importance in Chinese reading development ([55]). However, whether gross motor competencies similarly influence Chinese reading remains essentially unexplored, creating a significant gap in both theoretical models and educational practice.

### 1.5. Theoretical Rationale and Research Influences

The conceptual foundation for examining gross motor skills’ potential role in reading development derives from embodied cognition perspectives, which posit that those cognitive processes—including reading—are fundamentally grounded in bodily experiences and sensorimotor interactions. Within this framework, gross motor development may support reading through multiple pathways: enhancing postural control and classroom engagement, developing spatial awareness relevant to text processing, and fostering neural networks that subserve both motor coordination and cognitive functions.

Furthermore, several studies have reported non-significant or even negative associations between gross motor proficiency and reading abilities. For instance, [37] ([37]) identified a significant negative relationship between total motor proficiency and silent reading ability among Year 1 students, particularly noting that boys with higher gross motor proficiency tended to perform more poorly in reading. Similarly, [1] ([1]) observed no significant relation between motor skills (including gross motor tasks) and reading performance in their study. The relationship between upper limb coordination (a component of gross motor proficiency) and reading also failed to reach statistical significance in another investigation ([32]).

Beyond direct correlations, some research suggests that the relationship between motor coordination and academic achievement may be influenced by other factors, such as physical activity levels ([12]). Notably, sedentary behavior, often characterized by activities like reading or homework, has been positively associated with academic performance ([10]; [25]). Theoretical models also propose that motor coordination might exert an indirect influence on academic outcomes through cognitive processes. For example, [45] ([45]) demonstrated that the impact of motor coordination (specifically aiming and catching skills) on word reading and spelling was mediated by working memory. This suggests that gross motor skills may not directly facilitate reading ability but could contribute indirectly by supporting underlying cognitive capacities. The uncertainty surrounding the gross motor–reading link is further reflected in intervention studies. Evidence regarding the effects of physical activity interventions on reading remains less conclusive ([49]). For instance, a classroom-based gross motor program showed no definitive effect on reading outcomes ([33]), and another physically active academic intervention improved mathematics and spelling but found no difference in reading performance ([39]).

Given the empirical inconsistencies in existing research, coupled with the linguistic and cultural specificity of current evidence, the present study seeks to address two fundamental questions that remain inadequately resolved:What is the relationship between gross motor proficiency and reading abilities among Chinese primary school students?Does gross motor proficiency predict reading abilities among Chinese primary school students?

## 2. Method

### 2.1. Participants

A total of 107 Chinese students (Grade 1 to Grade 3, 63 boys and 44 girls) were recruited by simple random sampling from three urban elementary schools of Hebei, China, and their mean age was 8.70 years. All participants started their formal literacy instruction from Grade 1 and received education through Mandarin Chinese. All participants were typically developing children without diagnosed physical and cognitive disabilities. The ethical application was approved by the Faculty Ethics Committee of the authors’ institution. Informed consent in writing was obtained from participants’ parents or guardians prior to the data collection.

While the number of participants is 107, they have accurately represented the population of the sample schools in terms of proportion and quality. The time of data collection also allows an early process; otherwise, the participants will soon rise to higher grades. Given the developing nature of this study, we would like to conduct a cross-grade comparison under the guardians’ ethical permission, which also limits the volume.

### 2.2. Data Collection and Measures

The data has been collected between June and July 2025, before the end of each grade’s last semester. All participants were assessed by their Physical Education teachers or coaches who had received rigorous research training and instruction. To begin with, the test of gross motor proficiency was conducted with specialized equipment, such as exercise mats, balanced beams, and basketballs and administered at the gym or playground of their schools. Moreover, the other paper-and-pencil tests including non-verbal intelligence and reading abilities were completed in a group setting at participants’ regular classrooms.

#### 2.2.1. Non-Verbal Intelligence Test

Non-verbal intelligence tests highlight their significance in assessing cognitive abilities across diverse populations, often circumventing language barriers inherent in verbal assessments. [6] ([6]) emphasizes the development and standardization of non-verbal intelligence tests, underscoring their potential as reliable tools for measuring intelligence independent of language skills. This foundational work suggests that non-verbal tests can serve as effective alternatives or supplements to traditional verbal assessments, especially in populations with language differences. Their development, standardization, and application continue to enhance the capacity for equitable and accurate intelligence assessment, especially in contexts where verbal testing may be limited or inappropriate ([30]).

Raven Colored Progressive Matrices ([44]) is a standardized test to assess the nonverbal intelligence of children between 5 and 11 years old through abstract reasoning ([58]). The test comprises three sets with thirty-six items presented by incomplete colored-pictures, so participants should supplement the missing piece based on the given matrices and make correct choices. Within each set, items are arranged in increasing order of difficulty ([38]). There are 12 items taken from Group A of the Raven Colored Progressive Matrices in this task, with a sum of 12 points. The reliability of this task was *α* = 0.547.

#### 2.2.2. Gross Motor Proficiency Test

The Bruininks–Oseretsky Test of Motor Proficiency (BOT-2) Short Form is a valid and reliable standardized motor assessment tool, including gross and fine motor skills, used for individuals aged 4 to 21 years old ([7]; [32]; [37]). The section on gross motor proficiency contains three composites covering body coordination (subtest 1: bilateral coordination, subtest 2: balance), strength and agility (subtest 3: running speed and agility, subtest 4: strength), and manual coordination (subtest 5: upper-limb coordination). In subtest 1, bilateral coordination, participants are required to jump in place and tap their feet and fingers with upper and lower limbs being moved synchronously on the same sides. According to subtest 2, their balance ability is evaluated by walking forward on a line and standing on a balance beam with one leg. Then, they should finish the one-legged stationary hop to show their running speed and agility in subtest 3, and complete push-ups and sit-ups to display their strength in subtest 4. Finally, in subtest 5, their upper-limb skills are assessed by three activities with a basketball, including dropping and catching the ball, dribbling the ball, and throwing the ball at the target. The examiners, who are professional basketball coaches or physical education teachers, assessed participants’ gross motor proficiency according to the above five subtests with ten items in total and the corresponding scoring criteria. The total score is 58 points. The reliability of this task was *α* = 0.516.

#### 2.2.3. Reading Abilities Tests

Reading, a multifaceted cognitive process, is related to metalinguistic awareness, like phonological awareness ([11]; [48]) and morphological awareness ([18]; [61]), as well as vocabulary knowledge ([40]; [60]). All reading instruments were adapted and modified based on existing measurements and participants’ proficiency. Therefore, the reading abilities were measured by four researcher-designed tests, namely the Chinese Phonological Awareness Test, the Chinese Morphological Awareness Test, the Chinese Vocabulary Knowledge Test, and the Chinese Reading Comprehension Test.

Chinese Phonological Awareness Test contains four parts to evaluate participants’ tone awareness, onset-rime awareness, and phoneme awareness. In the tone awareness part, participants should identify the Chinese character whose tone is different from that of the other two, such as xīng (star), líng (bell), and hóng (red). The first character involves a high-level tone rather than a rising tone in the second and third ones, so it should be selected. In terms of two sections about onset-rime awareness, participants are supposed to identify the similarities and differences between two onsets or two rimes in a Chinese compound word with two characters. For instance, onsets in the compound word mèng xiǎng (dream) are different, but they are the same in yōu yǎ (elegance). Meanwhile, the compound word nán fāng (south) engages different rimes, while the word fēng shēng (the sound of wind) has the same ones. In the last part, the accuracy of phonemes should be judged in ten compound words with two to four Chinese characters, like the mistake in lù yè (green leaf), which should be lǜ yè (green leaf). There are 40 items with the sum of 40 points. The reliability of this task was *α* = 0.747.

Chinese Morphological Awareness Test contains two parts to assess participants’ morphological awareness of recognition and discrimination. In the first section, the relationship between two morphological constituents in Chinese compound words should be recognized. For example, there is an association between 鱼 (fish) & 金鱼 (golden fish), but such a link does not exist in the pair 花 (flower) & 花钱 (spend money) because they have different references. In the second part, participants need to discriminate homographic morphemes among three two-character compound words, such as 沙滩 (beach), 沙堡 (sand castle), and 沙发 (sofa). The morpheme 沙 in the first two words is related to *sand*, which is distinguished from that in the last one. There are 40 items with the sum of 40 points. The reliability of this task was *α* = 0.796.

Chinese Vocabulary Knowledge Test contains three parts to examine participants’ vocabulary breadth and depth. In the breath section, participants are required to match pictures and Chinese compound words accordingly, such as 眼镜 (glass), 窗户 (window), and 雨衣 (raincoat). In the depth part, the spelling and meaning of the words are checked by identifying the correct form of the constituent character, like 花丛 (flower bush) rather than 花从, or 蜜蜂 (bee) instead of 密蜂. Meanwhile, participants need to judge the meaning of words according to the context of sentences. For instance, in the sentence 这些安全标志, 有的很严肃, 有的很温馨. (Some of the safety signs are serious, while some are warm.), the selected word refers to the serious and formal content instead of serious facial expressions. There are 40 items with the sum of 40 points. The reliability of this task was *α* = 0.594.

Chinese Reading Comprehension Test contains two parts, namely, completing sentences based on the given pictures and answering sentence reading comprehension questions. To begin with, participants are expected to fill in the blanks of sentences based on the given pictures. Furthermore, ten sentences followed by ten questions, respectively, are presented in the second part, so participants should provide answers accordingly. There are 20 items in this task, with 1 point for each in the first section and 3 points for each in the next, and the sum is 40 points. The reliability of this task was α = 0.538.

Alpha is defined as the estimate of the reliability ([16]), which can be affected by the homogeneity or the heterogeneity of the scores of the tested population ([52]). It means that the score reliability can vary in different samples ([22]). The reliability of tests in this study was calculated based on the random sampling of participants and a set of items showed a high level of consistency in their scores (e.g., Item 1 and 10 in the Gross Motor Proficiency Test, Item 1, 2, and 3 in the Chinese Vocabulary Knowledge Test, and Item 1, 2, and 3 in the Chinese Reading Comprehension Test). Therefore, the reliability was negatively influenced to some extent. Nonetheless, all the items, except for Item 1 and 10, in the Gross Motor Proficiency Test were significantly and highly related to the sum score (*r* = 0.350–0.607, *p* < 0.05), demonstrating high internal consistency. Meanwhile, based on the correlational results, there were such strong and significant associations among subtests of reading abilities (*r* = 0.260–0.593, *p* < 0.01) that the robust construct validity was ensured.

## 3. Results

All the data underwent descriptive and inferential analysis. Firstly, the descriptive statistics displayed the fundamental results of measures. Secondly, the relationship among all variables was shown through the correlational analysis. Thirdly, the effect of gross motor proficiency on reading abilities was examined by the regression analysis, after controlling for age and non-verbal intelligence.

### 3.1. Descriptive Analysis

Table 1 shows the descriptive results of non-verbal intelligence, gross motor proficiency, and reading abilities. According to the skewness and kurtosis, all variables approximately followed a normal distribution, except for the control variable, non-verbal intelligence, and the constituent of gross motor proficiency, body coordination. Both variables exhibited a ceiling effect, with the average accuracy rates of 86.75% (non-verbal intelligence) and 98.53% (body coordination), respectively. The standard deviations indicated an adequate spread in the other variables.

### 3.2. Correlational Analysis

A correlation matrix in Table 2 shows the relationship patterns. Reading abilities were measured based on Chinese phonological awareness, Chinese morphological awareness, Chinese vocabulary knowledge, and Chinese reading comprehension, and these four variables showed significant correlations. Also, the effect sizes are representative (*r* = 0.260, *p* < 0.01 to *r* = 0.593, *p* < 0.01), the *r* values reflect a low-medium effect in the model, indicating that the correlations have various potential influence. Regarding the two control variables, age was significantly and negatively related to gross motor proficiency (*r* = −0.385, *p* < 0.01), but it was significantly and positively related to components of reading ability (*r* = 0.463, *p* < 0.01; *r* = 0.220, *p* < 0.05; *r* = 0.292, *p* < 0.01), except for Chinese reading comprehension. Non-verbal intelligence was only associated with Chinese morphological awareness (*r* = 0.212, *p* < 0.05) and Chinese reading comprehension (*r* = 0.210, *p* < 0.05). Importantly, no significant correlation was observed between gross motor proficiency and Chinese morphological awareness, nor between gross motor proficiency and Chinese reading comprehension. It even showed a weakly negative correlation with either Chinese phonological awareness (*r* = −0.228, *p* < 0.05) or Chinese vocabulary knowledge (*r* = −0.231, *p* < 0.05).

### 3.3. Regression Analyses

Pair plots in Figure 1 further present the weakly negative relationships between gross motor proficiency and reading subtests, which are consistent with the correlational results. Either phonological awareness or vocabulary knowledge demonstrated significant negative correlations with gross motor proficiency (*r* = −0.23, *p* < 0.05; *r* = −0.23, *p* < 0.05). There were no significant associations found between morphological awareness and gross motor proficiency, as well as reading comprehension and gross motor proficiency. Notably, compared to peers with high motor proficiency, participants exhibiting moderate-to-low motor levels showed superior performance in reading skills.

Follow-up regression analyses were conducted to investigate the contributions of gross motor proficiency to reading abilities, as assessed by phonological awareness, morphological awareness, vocabulary knowledge, and reading comprehension. Table 3 displays the hierarchical results after controlling for age and non-verbal intelligence. The effect sizes (*β* and $R^{2}$) in relevant studies have reached the common standard; this low-value trend may imply that complex factors have a hidden interaction, for example, social background. Thus, we may conclude that the result remains reliable and proceed to analyze it in the next section. As a result, gross motor proficiency accounted for only 0% to 1.6% variance in components of reading ability, justifying the negligible predicting power.

## 4. Discussion

The current study aimed to explore the relationships between gross motor proficiency and reading abilities among young Chinese primary school students. Generally, there were no significantly positive associations between these two variables, which was consistent with some previous findings ([1]; [12]; [37]; [50]). Meanwhile, gross motor level failed to predict reading achievements. Firstly, the results underscored the weakly negative relationship between gross motor proficiency and all reading subskills. Additionally, the negative relation between gross motor level and both morphological awareness and reading comprehension did not reach significance.

Although gross motor skills at kindergarten were significant in predicting cognitive achievement (mathematics and reading) during early school, their effect sizes were negligible ([50]). In the study conducted by [32] ([32]), upper limb coordination, the component of gross motor proficiency, was most strongly related to reading composite scores. Still, these relationships did not reach statistical significance. Likewise, a significant positive association between motor skills (gross motor skills like catching with one hand, throwing at a wall target, and shuttle run) and numeracy and a negative association between motor skills and the English test was observed in boys, but there was no significant relation between motor skills and reading ([1]). Bivariate correlations also revealed weak and insignificant associations between gross motor coordination and academic achievement scores (writing, reading, and mathematics performances). In addition, the relationship between motor coordination and academic achievement (writing performance) may be influenced by physical activity levels in boys ([12]). Based on the results of Structural Equation Modeling, motor coordination (especially aiming and catching skills) had an indirect impact on academic outcomes (word reading, spelling, and numerical operations) via working memory ([45]). Moreover, the evidence regarding the effects of physical activity interventions on reading was less conclusive ([49]), as also demonstrated by the classroom-based gross motor program on reading outcomes ([33]). Similarly, the systematic review showed that gross and fine motor skills were positively correlated with overall performance and language performance. However, the association between gross motor skills and students’ reading was uncertain ([55]). Physically active academic intervention significantly improved mathematics and spelling of elementary school children, but no difference was found in reading ([39]). Furthermore, [37] ([37]) justified a significant and moderate negative relationship between total motor proficiency and their ability to read silently with accuracy and fluency. A similar significant negative association also existed between gross motor proficiency (encompassing manual coordination, body coordination, and strength and agility) and reading ability. Notably, boys in Grade 1 with higher gross motor proficiency tended to perform more poorly in reading ability.

The negative relationships between gross motor proficiency and reading abilities among Chinese primary school students can be explained from cognitive and sociocultural frameworks. Drawing on a cognitive perspective, a series of seemingly unrelated behaviors share a common resource that is limited. It is noteworthy that ego depletion in one domain can result in impaired performance in another ([3]). Given this, high physical exertion like gross motor exercise may impair the cognitive resources for reading. Moreover, [15] ([15]) claimed that acute exercise could lead to the increased neurotransmitter concentrations, which may positively improve the speed of processing and negatively affect the accuracy of processing working memory tasks based on ([36]; [35]). Therefore, there is a possibility that gross motor activities can interfere with reading abilities relying on cognitive processing skills. Notably, according to the hypothesis of inverted-U effect of acute exercise on cognitive performance put forward by [17] ([17]), optimal cognitive performance would be achieved through moderate-intensity exercise, in contrast to the detrimental influence of low and high intensities ([35]).

From the sociocultural aspect, safe physical outdoor environments to move and explore freely are limited in China, in spite of their important role in gross motor development ([28]). Additionally, school-aged Chinese children, particularly those living in urban areas, are reported to have low physical activity and high sedentary behavior engagement ([53]; [54]; [63]), because a large amount of time is required for additional study and homework ([62]). Sedentary behavior is characterized by activities with energy expenditure less than 1.5 metabolic equivalents while in a sitting or reclining posture ([19]; [25]). The stronger the motor competence is in different abilities, such as kicking, throwing, and jumping, the greater the range of possibilities of physical activity practice that could replace sedentary behavior ([19]). Meanwhile, a low but positive correlation was found between gross motor skill scores and participation in physical activity ([20]). Likewise, children who had better gross motor skill performance spent significantly more time in either moderate-to-vigorous physical activity or vigorous physical activity and spent significantly less time in sedentary behaviors than peers with lower scores ([57]). However, a positive association existed between sedentary time, particularly non-screen time (reading or homework), and academic performance ([10]; [23]; [25]). It indicates that sedentary behavior may result in improved academic achievements like reading and decreased physical activities, which was negatively connected with gross motor competence in elementary school children.

## 5. Conclusions

A weak negative relationship was found between gross motor proficiency and reading abilities among Chinese primary school students. Meanwhile, there was a negligible predicting effect of gross motor level on reading achievements. On the one hand, high intensity of gross motor exercise may impair the cognitive performance, thus decreasing the accuracy of reading. On the other hand, this cohort of children tended to spend more time on sedentary behaviors like reading and homework, because of a huge academic burden and limited physical outdoor environments, and less energy was used for physical motor activities. Notably, educators should be aware that students with moderate-to-low motor proficiency perform better in reading, suggesting that curriculum design should maintain a balance between physical exercise and academic development. Also, family policy should consider students’ benefits across multiple aspects, suggesting that individual development is a dynamic process of multi-layer cognitive training.

The present study contains a few limitations. Firstly, the discipline requires an effective tool to consider additional covariates, such as school district, family income, and parental attitudes towards sports. To mitigate this, we introduced the social and cognitive theoretical mechanism in the discussion. In the future, methodology exploration should consider dynamic, systematic methods to enhance effect size and model reliability. Secondly, this study focuses on the general situation. Therefore, we may argue that it requires a case-crossover study and should include an additional qualitative investigation with extreme-case participants to explore further the relationship between gross motor proficiency and reading abilities.

## Figures and Tables

**Figure 1 behavsci-15-01613-f001:**
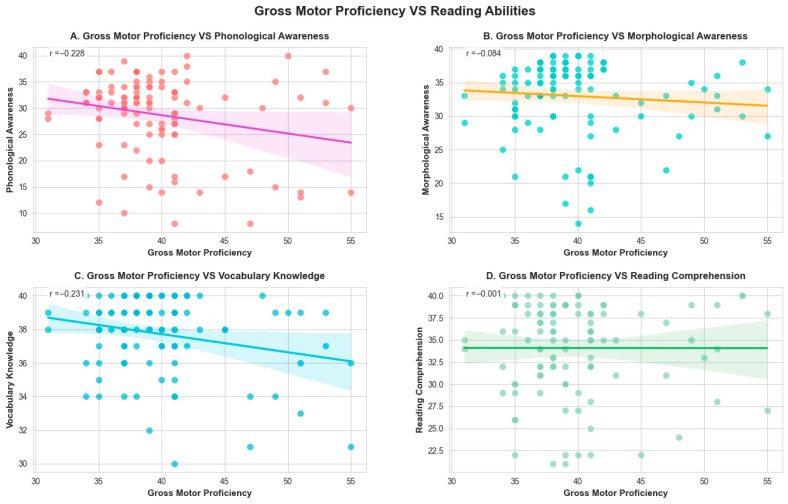
Gross Motor Proficiency and Reading Ability Relationships.

**Table 1 behavsci-15-01613-t001:** Descriptive statistics.

	Min	Max	M	SD	Skewness	Kurtosis
NVI (12)	2	12	10.43	1.561	−2.264	8.326
GMP (58)	31	55	39.82	4.870	1.334	1.783
BC (15)	11	15	14.78	0.756	−3.735	13.908
S&A (28)	9	26	13.85	3.392	1.984	3.525
MC (15)	7	15	11.20	1.896	−0.169	−0.836
RA						
CPA (40)	8	40	28.71	7.444	−1.103	0.534
CMA (40)	14	39	32.98	5.465	−1.496	1.939
CVK (40)	30	40	37.74	2.279	−1.263	1.303
CRC (40)	21	40	34.09	5.230	−0.917	0.012

NVI Non-verbal intelligence, GMP Gross-motor proficiency, BC Body coordination, S&A Strength and agility, MC Manual coordination, RA Reading ability, CPA Chinese phonological awareness, CMA Chinese morphological awareness, CVK Chinese vocabulary knowledge, CRC Chinese reading comprehension.

**Table 2 behavsci-15-01613-t002:** Bivariate correlations.

Variables	1	2	3	4	5	6	7
1. Age	1						
2. NVI	0.007	1					
3. GMP	−0.385 **	−0.028	1				
4. CPA	0.463 **	0.107	−0.228 *	1			
5. CMA	0.220 *	0.212 *	−0.084	0.347 **	1		
6. CVK	0.292 **	0.133	−0.231 *	0.536 **	0.580 **	1	
7. CRC	0.082	0.210 *	−0.001	0.260 **	0.593 **	0.403 **	1

NVI Non-verbal intelligence, GMP Gross-motor proficiency, CPA Chinese phonological awareness, CMA Chinese morphological awareness, CVK Chinese vocabulary knowledge, CRC Chinese reading comprehension. * *p* < 0.05, ** *p* < 0.01.

**Table 3 behavsci-15-01613-t003:** Hierarchical regressions predicting reading abilities.

	CPA		CMA		CVK		CRC	
	β	*t*	β	*t*	β	*t*	β	*t*
Age	0.441	4.698 **	0.221	2.177 *	0.238	2.377 *	0.096	0.928
NVI	0.103	1.187	0.211	2.246 *	0.127	1.374	0.211	2.194 *
GMP	−0.55	−0.589	0.007	0.071	−0.136	−1.352	0.042	0.403
$R^{2}$	0.228		0.093		0.118		0.052	
$∆R^{2}$	0.003		0.000		0.016		0.001	
SE	6.636		5.280		2.172		5.165	
∆F	0.346		0.005		1.828		0.163	

NVI Non-verbal intelligence, GMP Gross-motor proficiency, CPA Chinese phonological awareness, CMA Chinese morphological awareness, CVK Chinese vocabulary knowledge, CRC Chinese reading comprehension. * *p* < 0.05, ** *p* < 0.01.

## Data Availability

The raw data supporting the conclusions of this article will be made available by the authors on request.

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
