# Peer review of "Gross Motor Proficiency and Reading Abilities Among Chinese Primary School Students"

_behavsci, 2025, doi:10.3390/bs15121613_

Round 1
Reviewer 1 Report
Comments and Suggestions for Authors
Dear authors, there is much lacking in this manuscript for it to reach the scientific level it was aiming for. I fail to see the relevance of gross motor proficiency connection to reading abilities from the theoretical background you offerend in the text-why do you presume there is a connection, why would it be relevant, etc. since the connection is not well defined, the purpose of the research is not clear. You should explain why the Chinese language context of your research would contribute to the existing load of similar research among the alphabetical languages. The methodology of the paper lacks a decritption of how and when the data was collected, the scientific proof of researchers who conducted the research, the socio-demographic data on participants- was it collected/why/was it analysed/what are the results of those analysis...The instruments used-why was there a need to construct four new scales? The reliability of all the used sclaes was bellow the threshhold- very low reliability makes the result analysis close to irrelevant. Some of the conclusions are merely speculations ('high gross motor skills related to less homework writing and less reading exercises- no sound proof of such a conclusion). I believe you had good intentions to enrich the existing data on the motor skills and reading behaviour in a non-alphabetical language setting but there is much to be improved in this manuscript to reach that level.
Author Response
Dear Reviewer,
We sincerely appreciate the time and effort you have dedicated to providing constructive comments and valuable suggestions, which have been highly insightful in helping us improve the paper.
Attached herewith is a comprehensive response to your comments and concerns.
We eagerly anticipate your feedback and remain optimistic about the possibility of disseminating our research findings through publication in Behavioral Sciences.
We express our sincere appreciation for your invaluable guidance and thoughtful consideration.
Best regards,
Tongtong Shao, Feng Lu, Dingzhou Liu, Hongfan Chen and Haomin Zhang*

Reviewer 2 Report
Comments and Suggestions for Authors
Thank you for the opportunity to review the manuscript titled, Gross Motor Proficiency and Reading Abilities among Chinese Primary School Students. The manuscript explores an interesting and under-researched topic—the relationship between gross motor proficiency and reading abilities in Chinese primary school students. The study is clearly structured, includes appropriate statistical analyses, and references are up-to-date and relevant.
However, several areas require further clarification and strengthening before this paper can be accepted for publication. Below, I provide section by section feedback with the hope that it will help authors in improving the paper.
- Theoretical Context & Framing
-The introduction provides a good overview but would benefit from a clearer conceptual distinction between gross and fine motor skills in relation to reading development.
-The discussion of existing inconsistencies in the literature could be expanded to highlight possible theoretical mechanisms (e.g., embodied cognition, executive function links).
- 2. Methodology Clarifications
-Please justify the sample size (N=107) with a power analysis or rationale.
- The paper would benefit from a clearer explanation of the sampling strategy and data analysis procedures.
-The reliability scores (α values) of some instruments are low (i.e., α = 0.516, α = 0.538). Discuss how this may affect the interpretation of findings.
-Clarify whether socioeconomic background or parental education were considered or controlled, as they may confound reading outcomes. For example, specify how participants were selected and whether socio-economic or demographic variables were controlled.
- Results & Analysis
-The results are well presented but need stronger interpretative depth. For instance, the negative correlation between gross motor proficiency and reading could be contextualized with cultural or schooling practices in China (i.e.., academic pressure, limited physical activity time).
-Consider including effect sizes (β, R² change) explicitly in the text to strengthen transparency.
- Discussion and Implications
-The discussion would be more compelling if it connected findings to broader educational or developmental implications, i.e., balance between academic and physical education.
-The current discussion sometimes repeats results rather than providing critical interpretation. Add more synthesis and potential explanations.
- Language and Style
-The English is understandable but contains awkward phrasing and redundancies. A light professional proofreading is recommended.
-Pay attention to article usage, i.e., “the gross motor proficiency”, it should be: “gross motor proficiency”) and smoother paragraph transitions.
- Conclusion and Limitations
-The limitations section is concise but should explicitly address measurement reliability and cross-sectional design limitations (causality).
-Consider refining the conclusion to more explicitly summarize the educational implications of your findings for teachers and curriculum designers, and policy implications for balancing physical and cognitive activities in early schooling.
- References
-Standardize journal names (full names preferred).
-Correct minor typos (extra periods, line breaks).
-Include DOIs for all articles if available.
-Check consistency in volume/issue formatting.
-Emphasize recent (last 5–7 years) references if possible.
Overall, this is a valuable and well-structured paper that makes a moderate-to-strong contribution to literature. After minor revisions, it could be suitable for publication.
Comments on the Quality of English Language- Language and Style
-The English is understandable but contains awkward phrasing and redundancies. A light professional proofreading is recommended.
-Pay attention to article usage, i.e., “the gross motor proficiency”, it should be: “gross motor proficiency”) and smoother paragraph transitions.
Author Response

(The authors gave the same response as above.)

Reviewer 3 Report
Comments and Suggestions for Authors
I think this is a worthwhile study but there are some areas which I feel could be improved upon to make the overall paper feel more meaningful and robust, as follows:
- In the background section the literature review is quite dense and a little confusing in places so it doesn't form a clear narrative. In particular I think it could benefit from a clear reasoning as to why these research questions were being asked - how were the underlying hypotheses reached?
- In addition, you highlight that Mandarin Chinese is not an alphabetical language but it would be useful as a scholar to understand a little more about the language so that the assessments (which are described well, by the way) are more understandable to the reader.
- A more explicit explanation for inclusion of the Non-verbal Intelligence Test needs to be made.
- Some of the literature in the Discussion would benefit from being in the background section as it feels very relevant for the justification for the study as well as understanding the findings.
- In the Discussion and Conclusions you make some leaps in interpretation which I am not entirely sure are justified e.g. Ll328-332 ‘By replicating the results, the present study justified the same finding that Chinese young children with high gross motor proficiency showed stronger interest in motor exercise and used less time for sedentary activities like reading and homework, which in turn negatively affected their reading abilities.’ and ll336-339 ' This cohort of children tended to spend more time on sedentary behaviors like reading and homework, because of a huge academic burden and limited physical outdoor environments, and less energy was used for physical motor activities.' - how do you know this?
- Finely the English could do with a little bit of 'fine-tuning' for clarity.
Author Response

(The authors gave the same response as above.)

Round 2
Reviewer 1 Report
Comments and Suggestions for Authors
Dear authors, thank you for making all the requested modifications to the text. The paper still lacks firm methodological reasoning and postulations but I believe it can not be improved more than this. Other researcher who will base their research on this one can work on the improvement of the instrument.
Author Response
Dear Reviewer,
Thank you for your positive feedback and for acknowledging our modifications. We fully understand and agree with your point regarding the methodological limitation. It was also highlighted as a direction for future research in the conclusion.
Page 11, Line 420–424:
Firstly, the discipline requires an effective tool to consider additional covariates, such as school district, family income, and parental attitudes towards sports. In the future, methodology exploration should consider dynamic, systematic methods to enhance effect size and model reliability.
We believe this strengthens the paper’s contribution by clearly framing its findings as a foundation for further inquiry. Thank you again for your insightful guidance.
Reviewer 3 Report
Comments and Suggestions for Authors
Many thanks for revising your manuscript and thoughtful responses. There are some minor English tweaks that could be made but otherwise I am happy with the changes you have made e.g.:
l11 ‘the existing studies’ delete ‘the’
l14-15 the tests should be ‘a’ rather than ‘the’
;19 ‘achievements’ should be ‘achievement’
l144 ‘are simple random sampling’ unclear what you mean here
l264 ‘the high internal consistency’ delete ‘the’
l387 ‘were limited’ change to ‘are limited’?
ll386-391 the tense needs to be changed e.g. ‘were reported’ changed to ‘are reported’
ll417-418 ‘the family policy’ change to ‘family policy’
Author Response
|
Dear Reviewer, We are so grateful and happy for your acknowledgment of our revisions. We have carefully addressed all the suggested English tweaks. All changes made in response to your comments have been highlighted in blue within the attached manuscript. |
